# The Influence of Probiotic Supplementation on the Severity of Anxiety and Depressive Symptoms; Function and Composition of Gut Microbiota; and Metabolic, Inflammation, and Oxidative Stress Markers in Patients with Depression—A Study Protocol

**DOI:** 10.3390/metabo13020182

**Published:** 2023-01-25

**Authors:** Anna Skowrońska, Oliwia Gawlik-Kotelnicka, Aleksandra Margulska, Dominik Strzelecki

**Affiliations:** 1Department of Affective and Psychotic Disorders, Central Teaching Hospital, Medical University of Łódź, ul. Czechosłowacka 8/10, 92-216 Łódź, Poland; 2Department of Adolescent Psychiatry, Central Teaching Hospital, Medical University of Łódź, ul. Czechosłowacka 8/10, 92-216 Łódź, Poland

**Keywords:** microbiome, metabolism, depression, anxiety disorders

## Abstract

This article aims to present the theoretical basis, methodology, and design of a clinical trial we will conduct. The study will be prospective, randomized, placebo-controlled, and double-blind. Each intervention period will last 8 weeks and the trial will be conducted on 100 patients in total, who will be randomly divided into two groups consisting of 50 patients each. We plan to investigate the impact of *Lactobacillus helveticus Rosell* and *Bifidobacterium longum Rosell* on the depressive, anxiety, and stress levels in patients with depressive disorders with possible comorbid anxiety. In addition to assessing the influence of probiotics on the clinical condition, we also plan to study the clinical and biochemical parameters of metabolic syndrome, which often coexists with depression. Both depressive and metabolic issues may have part of their etiopathology in common, e.g., inflammation, oxidative stress, and dysbiosis. This is why we will additionally investigate the parameters related to gut microbiota, inflammatory, and oxidative statuses. Thus, the primary endpoint of the study will be the change in depression score measured with the Montgomery–Åsberg Depression Rating Scale. The secondary endpoints will include changes in anxiety and stress levels, as well as metabolic, inflammation, and oxidative stress parameters.

## 1. Introduction

Depression is one of the most common and debilitating disorders worldwide [1]. Moreover, depressive disorders often coexist with somatic diseases, further increasing mortality risks [2,3]. Recent studies uncovered that depression and other civilization diseases are associated with chronic, low-grade inflammation. This is characterized by increased circulating pro-inflammatory cytokines, altered leukocyte population frequencies in blood, and the accumulation of activated immune cells in tissues, including the brain [4]. Although far from conclusive, emerging evidence suggests that chronic inflammatory conditions in the central and peripheral immune systems may mediate a subset of depressive disorders [5] that are commonly concurrent with obesity. Several studies showed that oxidative stress (the imbalance between the production of free radicals and their neutralization by antioxidants), along with inflammation, leads to the development of depression and its comorbidities [6,7,8,9,10].

Elucidating these mechanisms linking depression, inflammation, and oxidative stress could generate potential new therapeutic targets or patient-specific strategies to combat both depressive disorders and their coexisting conditions.

Recently, there has been much interest in the role of intestinal microbiota changes in the pathophysiology of civilization diseases. The term “microbiota” refers to all microorganisms in a specific environment [11]. The intestines are where the microbiota population is most abundant in the human body [11]. Its basic functions include maintaining the integrity of the intestinal wall, protection against pathogenic bacteria and viruses, and the synthesis of key vitamins [11,12]. In addition, microbiota participates in the metabolism of nutrients, e.g., converts dietary carbohydrates into short-chain fatty acids (SCAFs)—namely, butyrate, acetate, and propionate—which are the most representative metabolites of anaerobic intestinal fermentation of fiber, and assessments of their level are used to analyze the function of the microbiome [13]. It was shown that the levels of SCFAs and the ratio of their three main components can also be good markers of intestinal dysbiosis and serve as a reliable approach to the assessment of microflora function [14]. There is more and more evidence showing that aberrant gut microbiota may lead to chronic inflammation [15,16] and oxidative stress exacerbation in tissues [17] and may serve as a link between depression and dysbiosis.

Experimental studies conducted in animal models over the past few years confirmed the relationship between dysbiosis (imbalanced microbiome) and depression-like behavior [18,19,20,21]. Similar relationships were observed in human studies. The results of the research revealed a clear diversification of the intestinal microflora (different types and species of bacteria) [22,23,24]. On top of that, patients diagnosed with major depressive disorder (MDD) show altered levels of butyrate, acetate, and propionate [25,26]. In the patients with higher amounts of butyrate-producing bacteria, such as *Faecali bacterium* and *Coprococcus* spp., the parameters of quality of life were more favorable [27]. It was also found that propionate administration alleviates depressive symptoms in mice [27]. The composition of the analyzed fecal microbiota, typically found in patients with depressive disorders, is heterogenous [22,23,24]. The functional level of the gut ecosystem is mostly independent of the taxonomic order, and a taxonomic approach alone is unlikely to be sufficient to reliably assess the health and function of the microbiota [28]. Thus, microbiota composition and fecal SCFA levels will be investigated in the study. 

The administration of probiotics (live microorganisms intended to provide health benefits to the host when administered) is one of the methods of modifying disturbed microbiome imbalance [29]. Recent studies, including two meta-analytical summaries, emphasized the beneficial role of probiotic supplementation and the reduction in the severity of depressive symptoms in patients with mild or moderate depressive episodes [30,31]. However, the number of studies that assessed the role of probiotic supplementation in patients with the above diagnosis remains insufficient and of various designs. More research on patients with anxiety and depression symptoms is necessary, as most of the interventions were carried out on mentally healthy individuals.

What is particularly interesting for us in the context of the project. Some studies were focused on assessing the potential impact of probiotic supplementation on the symptoms of anxiety coexisting with depression [32,33]. However, these studies yielded conflicting results. In some of them, probiotic supplementation (lasting from 8 to 12 weeks) reduced anxiety symptoms, while in other studies and two meta-analytical summaries, probiotic supplementation did not significantly affect anxiety symptoms in humans, but decreased their severity in animals [34,35]. Bearing in mind the inconclusive results of previous research, we plan to investigate these issues in depth. Furthermore, depression with anxiety features and mixed depressive and anxiety disorder (MDAD) are now integral parts of the depressive disorders section of the International Classification of Diseases (ICD-11).

According to recent research, probiotic supplementation, which provides a reduction in intestinal dysbiosis with immunomodulatory and anti-inflammatory effects, may also help to treat metabolic abnormalities. The latest meta-analysis of probiotic supplementation’s impact on metabolic syndrome parameters indicated an improvement in the body mass index (BMI), lipid profile, and glucose metabolism [36,37]. Moreover, probiotic supplementation may have a positive effect on the parameters of inflammation and oxidative stress [38,39]. There are only a few studies on this subject available, in which malondialdehyde’s (MDA) serum level was significantly decreased, while the total antioxidant capacity (TAC) remained unchanged [40] during probiotics supplementation. Malondialdehyde is one of the final products of polyunsaturated fatty acid peroxidation in the cells and has a mutagenic and carcinogenic effect [41]. Total antioxidant capacity (TAC) is a popular diagnostic test that is frequently used to assess the antioxidant status of biological samples and can estimate the antioxidant response against the free radicals produced in the organism. Thus, we intend to observe some of the metabolic, inflammatory, and oxidative parameters during probiotics supplementation; these issues have not been studied thoroughly so far. It may help to distinguish the subpopulation of depressed subjects that responds positively to probiotic supplementation, as well as some biomarkers of probiotics efficacy in depression.

The aim of this study was to gather empirical evidence that will enable us to evaluate the impact of probiotic supplementation on the severity of anxiety and depressive symptoms, function and composition of gut microbiota, metabolic parameters, and inflammation and oxidative stress markers in patients with diagnosed depressive disorders.

Our scientific hypotheses were the following:-Comorbidity of anxiety symptoms in patients with diagnosed depressive disorders increases the likelihood of therapeutic response to probiotic supplementation in both depressive and anxiety symptoms;-Metabolic abnormalities, chronic inflammation, and oxidative stress markers may be predictive of probiotic efficacy toward depression;-Probiotic supplementation increases the concentration of fecal short-chain fatty acids and the level of diversity of gut microbiota composition in depressed subjects;-Probiotic supplementation improves metabolic parameters in patients with depressive disorders;-Probiotic supplementation improves inflammation and oxidative stress parameters in patients with depressive disorders;-Functional biomarkers (the ratio of levels of short-chain fatty acids) but not classification (taxonomic) changes of microbiota are typical for depressive disorders.

In our trial, the primary outcome measure will be the severity of depressive symptoms assessed with the Montgomery–Åsberg Depression Rating Scale (MADRS). The secondary outcome measures will include depression, anxiety, and stress symptoms assessed with the Depression, Anxiety, and Stress Scale (DASS) with subscale scores; the quality of life level assessed with the WHOQOL-BREF instrument; blood pressure (BP), body mass index (BMI), and waist circumference (WC) measures; and fasting glucose (fGlc), HDL cholesterol (HDL-C), triglycerides (TG), white blood cell count (WBC), neutrofiles, serum levels of C-reactive protein (CRP), levels of fecal SCFAs, fecal microbiota α-diversity, and the levels of oxidative stress parameters (total antioxidant capacity (TAC) and malondialdehyde (MDA)) in the blood serum (Table 1).

## 2. Materials and Methods

### 2.1. Scale and Questionnaires

Study questionnaire (SQ)—this will be used to gather basic information on sociodemographics and data related to health. The participants will be asked to give information on their personally identifying data, diet, nicotine smoking, physical activity, education, living place, professional activity, and whether they are taking dietary supplements and drugs. The personal data will be pseudonymized.

Montgomery–Åsberg Depression Rating Scale (MADRS)—a 10-item, well-validated diagnostic questionnaire used to measure the severity of depressive episodes in patients with mood disturbances. The questionnaire includes questions on the following symptoms: (1) apparent sadness, (2) reported sadness, (3) inner tension, (4) reduced sleep, (5) reduced appetite, (6) concentration difficulties, (7) lassitude, (8) inability to feel, (9) pessimistic thoughts, and (10) suicidal thoughts [42,43].

The 21-item Depression, Anxiety, and Stress Scale (DASS-21)—a 21-item self-administered questionnaire that was designed to measure the magnitude of three negative emotional states: depression, anxiety, and stress. It is assumed that anxiety and stress are two states and can be analyzed separately. Each of the three DASS scales contains 7 items, divided into subscales of 2–5 items with similar content. The depression scale assesses dysphoria, hopelessness, devaluation of life, self-deprecation, lack of interest/involvement, anhedonia, and inertia. The anxiety scale assesses autonomic arousal, skeletal muscle effects, situational anxiety, and subjective experience of anxious affect. The stress scale is sensitive to levels of chronic non-specific arousal. It assesses difficulty relaxing, nervous arousal, being easily upset/agitated, irritable/over-reactive, and impatience. Subjects are asked to use a 4-point severity/frequency scale to rate the extent to which they have experienced each state over the past week. The scores for depression, anxiety, and stress are calculated by summing the scores for the relevant items [44].

WHO Quality of Life-BREF Scale (WHOQOL-BREF)—a well-known 26-item questionnaire that evaluates four domains: physical health, psychological health, social relationships, and environment [45].

Food Frequency Questionnaire (FFQ-6)—a questionnaire designed to assess the consumption of different food groups. Patients are asked to report their “normal consumption” over the past year. We plan to use the Polish self-administered Food Frequency Questionnaire 6 (FFQ-6) by Wądołowska [46].

International Classification of Diseases ICD-11 research criteria for 6A70-73, 6A7Y, 6A7Z [47]—we decided to include the whole of the new category of depressive disorders as the first stage of the eligibility screen in our project (Table 2).

### 2.2. Biological Parameters

Physical examination parameters: blood pressure (BP), weight, body mass index (BMI), and waist circumference (WC). 

We decided to measure waist circumference because central obesity is defined as waist circumference ≥ 94 cm for Europid men and ≥80 cm for Europid women, with ethnicity-specific values for other groups, and it is one of the components of metabolic syndrome diagnosed according to the International Diabetes Federation (IDF) criteria [48].

Blood:

Venous blood will be collected by qualified nurses. The collection will be performed under conditions of fasting after overnight resting in the morning between 8:00 and 10:00 a.m. A complete blood count with differential levels of C-reactive protein, HDL cholesterol, triglycerides, and fasting glucose will be measured in blood serum. The level of oxidative stress will be identified as a ratio of the total antioxidant capacity (TAC) and the level of malonedialdehyde (MDA) will be measured in blood plasma.

To obtain serum for future analyses, the blood will be centrifuged at 1500× *g* for 10 min. The blood samples will be stored in Eppendorf tubes and frozen at a temperature of −80 °C. After utilizing the blood in the experiments, it will be properly disposed of.

Feces:

Each patient included in the study will be asked to collect a sample of feces twice in a specially designed double-sealed bag, which will allow for keeping anaerobic conditions. The subjects will be required not to use laxatives, including mineral and castor oil, and not to consume synthetic fat substitutes or dietary supplements that block fats. It is advisable to wait for 48 h after the administration of a barium enema. The patients will collect stool samples after a night’s rest and deliver them to the investigators. The samples will then be divided into smaller ones and stored at a temperature of −80 °C until further analyses. Fecal short-chain fatty acids (SCFAs), including acetic, propionic, and butyric acids, will be measured with the liquid chromatography–mass spectrometry (LC-MS) technique. To detect potential changes in the composition of microorganisms, we will use 16S ribosomal RNA gene sequencing with a specification of variable regions 3 and 4 (V3–V4). The procedure will be carried out in four stages: (1) DNA isolation involving a qualitative and quantitative evaluation, (2) preparing DNA 16S (V3–V4) library with validation, (3) new-generation sequencing (NGS) on an Illumina 16R V3–V4 rRNA platform with paired-end 2 × 250 pz with fastq files generation, and (4) bioinformatic analysis and preparing a report. Sequences of 16S ribosomal RNA genes will be grouped in operational taxonomic units (OTU) at a similarity cut-off value equal to 97%. 

### 2.3. Ethics

The study will be conducted in compliance with the Declaration of Helsinki. The principal study investigator (PSI) has obtained the consent of the Bioethics Committee of the Medical University of Łódź for the whole protocol (resolution of the Bioethics Committee of the Medical University of Łódź, number RNN/109/20/KE of 28 April 2020). Each participant of the study will receive information on the design and aim of the study in writing. The patient will be included in the study if they meet the inclusion criteria and after they give informed written consent for such an inclusion. The subjects will be informed about the possibility of withdrawing from the study at any stage. There is no risk of negative consequences involved in participation in the study.

### 2.4. Statistical Analysis

Regarding the microbiota composition, we will assess the α-diversity (within-sample diversity and richness using the observed OTUs), β-diversity (diversity of microbial community structure), and taxa abundance. Associations between clinical symptoms severity (according to MADRS and DASS scores) and α-diversity indexes, as well as β-diversity dissimilarities, will be tested using permutational ANOVA (PERMANOVA). Associations between symptoms severity and relative abundances of phyla, genera, and species will be tested using multivariate associations. Models will be multi-adjusted for the covariates: age, dietary variables, BMI, and other confounding factors. The level of significance for statistical tests will be 5%. All statistical analyses will be performed using Statistica 13.3 (StatSoft, Tulsa, OK, USA). 

#### Sample Size Calculation

As for the primary outcome of the study, namely, the MADRS score, we will use a standard deviation of 5.15 and a difference in the mean of 2.9 (M1 = 20.0, M2 = 17.1). We based our assumptions regarding the mean and standard deviation of MADRS in a depressed population on a study by Dijkstra et al. [49]. A difference in the MADRS of 2.2 points or higher seems clinically relevant (Cohen’s D has a value of 0.4, while a value of 0.4 or higher is widely considered to be associated with a clinically relevant effect size). Using an online tool (https://www.stat.ubc.ca/~rollin/stats/ssize/n2.html, accessed on 2 December 2022), we calculated the required sample size (assuming that both groups taking part in the study are the same size). The total sample size for this study should be around 100 (M1 = 20.0, M2 = 17.1, SD = 5.15, alpha = 0.05, power = 80%).

More details of sample size calculations for different secondary outcome measures are described in Appendix A.

## 3. Results

As a result of our investigation, we created a study design, which is presented below.

### 3.1. Patients

Study groups:

Group I “PRO-D”—patients with diagnosed depressive disorders; meeting the criteria of the International Classification of Diseases (ICD-11) for 6A70-73, 6A7Y, and 6A7Z; and taking a probiotic composed of two bacteria strains: *Lactobacillus helveticus Rosell* and *Bifidobacterium longum Rosell*. The severity of depression will be evaluated with the use of the Montgomery–Åsberg Depression Rating Scale (MADRS). The patient will be included in the group after obtaining 13 or more points.

Group II “PLC-D”—patients with diagnosed depressive disorders; meeting the criteria of the International Classification of Diseases (ICD-11) for 6A70-73, 6A7Y, and 6A7Z; and taking a placebo. The severity of depression will be evaluated similarly as in group I and patients will be enrolled after obtaining 13 or more points in the MADRS. In both groups, the authors will measure the patients’ severity of anxiety symptoms with the Depression, Anxiety, and Stress Scale (DASS) to further analyze patients according to their level of anxiety and stress.

The designed study will be prospective, randomized, placebo-controlled, and double-blind. The intervention period will last 8 weeks for each patient and the study will be conducted on 100 patients in total, who will be randomly divided into two groups, consisting of 50 patients each.

#### Randomization

The participants will undergo randomization and will be randomly allocated into two groups (probiotic or placebo treatment) in a 1:1 ratio by an independent researcher. Neither the subjects nor the investigators will be aware of the treatment assignment until the end of the trial. Randomization will be computed using a computer-based random number generator (https://www.randomizer.org/, accessed on 8 March 2021). The distribution of the pills will also be done by an independent researcher. The researchers agreed that unblinding should occur for the patient’s safety only if any serious adverse events occur during the investigation.

### 3.2. Design

The patients from the two clinical groups will undergo pharmacological and psychological treatment in compliance with the procedures of the clinic. It is worth mentioning that the researchers will at no stage interfere in the treatment process.

The patients will be recruited in the Central Teaching Hospital (inpatient and outpatient clinics), Medical University of Lodz, Poland.

We plan to measure the following metabolic parameters: blood pressure (BP), weight, body mass index (BMI), and waist circumference (WC). Blood samples of each patient will be collected to determine serum CBC and measure the levels of C-reactive protein, HDL cholesterol, triglycerides, and fasting glucose. Microbiota function (as SCFA concentrations in feces) and composition will be assessed. The oxidative stress level in the blood will be assessed as the total antioxidant capacity (TAC) and the level of malondialdehyde (MDA) concentration in blood plasma.

The patients will be observed for twenty weeks and they will be administered probiotic supplements for the first eight weeks.

The study will consist of the following visits:

V0 “recruitment visit”: assessment of inclusion and exclusion criteria, giving informed consent, a full psychiatric and somatic examination, filling in a survey questionnaire (SQ), and filling in the MADRS scale;

V1 “randomisation visit” (within 5 days following V0): filling in the DASS and WHOQOL-BREF scales; measuring BP, WC, and BMI; blood and stool collection; and • t1-t3:3/8 monitoring using personal contact via phone or e-mail every two weeks; 

V2 “final visit” after 8 weeks following V1: filling in MADRS, DASS, and WHOQOL-BREF scales; measuring BP, WC, and BMI; and blood and stool collection; 

V3 “follow-up visit” after 12 weeks following V2: filling in the SQ, MADRS, DASS, and WHOQOL-BREF scales and measuring BP, WC, and BMI.

#### Eligibility Criteria (Table 3)

To be eligible for the trial, subjects must fulfill all the inclusion criteria and none of the exclusion criteria, as stated in Table 3.

**Table 3 metabolites-13-00182-t003:** Eligibility criteria. BMI: body mass index; GFR: glomerular filtration rate; ICD-11: International Classification of Diseases 11th Revision; MADRS: Montgomery–Åsberg Depression Rating Scale; NSAIDs: nonsteroidal anti-inflammatory drugs; PPIs: proton-pump inhibitors; TASR: Tool of Assessment of Suicide Risk; TSH: thyroid stimulating hormone.

Inclusion Criteria	Exclusion Criteria	Reasons for the Participant to Be Discontinued from the Study
1. Depressive disorders diagnosed according to the ICD-11; 2. Age between 18–70 years; 3. MADRS score > = 13; 4. Anti-depressant and anti-anxiety medications not changed within 3 weeks prior to the recruitment visit.	1. Pregnancy; 2. An infection/vaccination and/or treatment with antibiotics in the previous 4 weeks; 3. Supplementation with pro- or prebiotics in the previous 4 weeks; 4. Having a diagnosis of an autoimmune disease, being seriously immunocompromised, inflammatory bowel disease, cancer, IgE-dependent allergy, or severe kidney failure in the previous 4 weeks; 5. BMI > 35; 6. GFR < 30 mL/min/1.72 m^2^; 7. Unstable thyroid dysfunction (TSH < 0.27 or >4.2 µIU/mL) in the previous 4 weeks; 8. Psychiatric comorbidities (except specific personality disorder, additional specific anxiety disorder, and caffeine and nicotine addiction); 9. Regular treatment (more than 3 days a week) with PPIs, metformin, laxatives, systemic steroids, or NSAIDs in the previous 4 weeks; 10. Significant change in dietary pattern in the previous 4 weeks; 11. Significant change in daily physical activity or undertaking an extreme sports activity in the previous 4 weeks; 12. Significant change in dietary supplementation in the previous 4 weeks; 13. Significant change in smoking pattern in the previous 4 weeks; 14. High risk of suicide; 15. Is participating in, or has recently participated in, another research study involving an intervention that may alter the outcomes of interest for this study; 16. Any other condition or situation which, in the view of investigators, would affect the compliance or safety of the individual taking part.	1. Withdrawal of informed consent; 2. An infection/vaccination and/or treatment with antibiotics during the trial; 3. Consuming any probiotics other than those studied during the trial; 4. Lack of compliance with the probiotic supplementation; 5. Any change in the drug regimen during the study; 6. Exclusion criteria found after enrolment; 7. Any serious adverse event during the trial.

### 3.3. Interventions

At the beginning (V0), the patients will be asked to perform routine physical activity and not to modify their diet during the study. Patients included in group I (PRO-D) will receive one capsule daily containing a probiotic mixture at a daily dose of 3 × 10^9^ colony-forming units (CFU). The probiotic will be composed of two bacteria strains (*Lactobacillus helveticus Rosell^®^-52* and *Bifidobacterium longum Rosell^®^-175*), excipients (potato starch and magnesium stearate), and the capsule shell (made of hydroxypropylmethylcellulose). It will be ordered directly from the manufacturer (Sanprobi Stress^®^, Sanprobi Sp. z o. o., Sp. k., Szczecin, Poland). In our study protocol, we decided to use the combination of these two strains because both strains were studied in experimental and animal models and were found to improve emotional behavior in animals [50,51,52]. Clinical studies with healthy subjects and depressed patients showed inconclusive results [50,53,54,55,56]. We created a summative table with the list of the recent clinical trials with the efficacy of *Bifidobacterium longum* and *Lactobacillus helveticus* toward depressive, anxiety, and stress symptoms that could give the rationale for the combination of these two strains (Table 4). Interestingly, these two strains seem to also have specific effects on pro-inflammatory cytokine synthesis by reducing the level of pro-inflammatory cytokines, such as IL-1β, IL-6, and TNF-α, which may aid their antidepressant effect [57,58]. Moreover, recent study results suggested that Bifidobacterium species (spp.) positively correlated with both improved insulin resistance and obesity, and this might be associated with metabolic syndrome [59,60].

Patients included in group II (PLC-D) will receive the same capsule daily, containing only the excipients (potato starch and maltodextrin) and the capsule shell. The color, smell, and taste of the placebo will not be different from those included in the probiotic capsule (manufacturer—Sanprobi Sp. z o. o., Sp. k., Szczecin, Poland). The pill boxes will be labeled the same way. We decided that the intervention period will last 8 weeks because the recent clinical trials using probiotics demonstrated that it is rational to supplement the probiotics for at least 6–8 weeks to reduce the depression scale score [61,62,63]. We will assess how compliance is with personal, telephone, or e-mail contact every 2 weeks according to the monitoring questionnaire (MQ). Participants will be asked to collect the blister packs to evaluate compliance after the study and to fill in a daily medication chart. Patients will be considered compliant if they consume > = 80% of the supplements.

## 4. Discussion

Anxiety and depressive symptoms frequently co-occur. This fact may result in worsened functioning, a poorer prognosis, and a more severe and more chronic course of depressive disorders [64]. Unfortunately, although many joint studies were conducted, the common etiology of anxiety and depressive symptoms is still partly unclear. 

Consideration of the clinical picture is essential, as it allows for the identification of specific biomarkers and elucidation of the pathophysiological processes underlying the co-occurrence of depression and anxiety. In clinical practice, patients with such a clinical manifestation are often a major therapeutic challenge, as they less frequently respond to standard antidepressant treatment [65]. Recent studies, including meta-analytical summaries, emphasized the beneficial role of probiotic supplementation in reducing depressive symptoms when administered as an add-on [66]. Moreover, there is some evidence of the efficacy of probiotics toward symptoms of anxiety and stress [32,33,34]. Additionally, studies on new therapies confirmed that some of the strains of probiotics can also improve the clinical and laboratory components of metabolic syndrome and decrease the level of inflammation and oxidative imbalance [67]. Therefore, metabolic, inflammatory, or oxidative parameters may potentially serve as biomarkers of the therapeutic efficacy of probiotics toward depression with/without anxiety and stress features.

Clearly, our study protocol has both strengths and limitations. First of all, we decided to use the most recent classifications, i.e., ICD-11, in our study protocol. According to the ICD-11, depressive disorders include single-episode or recurrent depressive disorder, mixed depressive and anxiety disorder (MDAD), and dysthymia [68]. The distinction of depression with comorbid anxiety symptoms (“anxious depression”) underlying its clinical significance has an impact on a patient’s everyday functioning and quality of life [28]. Moreover, it is connected with its importance in primary care settings and because of evidence of its overlap with mood symptomatology [69]. Few studies investigated potential new treatment options for patients with MDAD and there is a huge need in the field. Therefore, we decided to include the whole category of depressive disorders as the first stage of the eligibility screen in our study. On the other hand, the study population will not be completely homogenous, as it will include both the category of mixed depressive and anxiety disorder (MDAD) and the category of major depressive disorder (MDD). 

It is well known that many factors that influence our microbiome, such as diet, treatment with antibiotics, metformin, proton pump inhibitors (IPP), viral or bacterial infections, vaccinations, air pollution, and cigarette smoking [70,71,72,73]. Obviously, we included the factors that may influence the microbiota changes in our microbiome in our original questionnaires, such as the survey questionnaire (SQ) at the beginning of the protocol and in the monitoring process every two weeks according to the monitoring questionnaire (MQ). However, we are not able to control for some factors, such as stress level, air pollution, or genetics.

Another limitation can be an insufficiently large patient population; the study will be conducted on 100 patients in total. Although the sample size in our study enables us to detect statistical significance, the authors are aware that it might not be enough for achieving clinical significance. In the case of research on microbiota influencing depressive symptoms, the reduction in depressive symptoms according to various scales is usually by a small number of points [56,74,75]. Moreover, a smaller sample size might in some cases fail to reject a null hypothesis. Therefore, further studies should be carried out. On the other hand, the calculated sample size seems similar to those from other studies regarding the effects of microbiota in psychiatric conditions [53,54,55].

In recent years, there has been a strong case for implementing a holistic approach to depression and recognizing it as a systemic disorder. The theory regarding the gut microbiome’s influence on the etiopathogenesis of depressive disorders is consistent with this approach [29,30,31,32], which confirms that probiotic supplementation may reduce depressive and anxiety symptoms. However, the number of such studies is still insufficient and their results are inconclusive. Studies on clinical populations are needed, as most interventions were conducted on healthy individuals. According to the most recent knowledge, probiotic supplementation may play an important role in modifying metabolic parameters [35] and favorably affecting inflammatory and oxidative stress parameters [37]. Our long-term scientific goals include understanding the relationship between the gut microbiota and anxiety and depressive symptoms, identifying a group of patients in whom probiotic supplementation may be an important component of antidepressant treatment, and generating a broad spectrum of therapeutic options for use in the clinical management of anxiety and depressive disorders. 

## Figures and Tables

**Table 1 metabolites-13-00182-t001:** Primary and secondary outcome measures. BMI: body mass index; BP: blood pressure; CRP: C-reactive protein; DASS: Depression, Anxiety, and Stress Scale; fGlc: fasting glucose; HDL-c: high-density protein cholesterol; MADRS: Montgomery–Åsberg Depression Rating Scale; MC: microbiota composition; MDA: malondialdehyde; SCFAs: short-chain fatty acids; TAC: total antioxidant capacity; TG: triglycerides; WBC: white blood cells; WC: waist circumference; WHOQOL-BREF: The World Health Organization Quality of Life-BREF.

	Materials and Methods			
	Psychometric Tools	Biological Samples		Physical Examination
		Blood	Feces	
Primary outcome measures				
Depressive symptoms	MADRS			
Secondary outcome measures				
Anxiety symptoms	DASS			
Stress level	DASS			
Quality of life	WHOQOL-BREF			
Metabolic parameters		fGlc, HDL-C, TG		BP, BMI, WC
Microbiota function			MC, SCFAs	
Inflammations parameters		WBC, NEU, CRP		
Oxidative stress parameters		TAC, MDA		

**Table 2 metabolites-13-00182-t002:** Depressive disorders in the ICD 11.

Depressive Disorders in the ICD 11	Code
Single-episode depressive disorder	6A70
Recurrent depressive disorder	6A71
Dysthymic disorder	6A72
Mixed depressive and anxiety disorder	6A73
Other specified depressive disorders	6A7Y
Depressive disorders, unspecified	6A7Z

**Table 4 metabolites-13-00182-t004:** The list of the recent clinical trials that demonstrated the efficacy of *Bifidobacterium longum* and *Lactobacillus helveticus* toward depressive, anxiety, and stress symptoms.

The List of the Recent Clinical Trials That Demonstrated the Efficacy of *Bifidobacterium longum* and *Lactobacillus helveticus* toward Depressive, Anxiety, and Stress Symptoms				
The Authors	Animal/Human Studies	Duration	Population	Results
Kazemi et al., 2018 [53]	Humans	8 weeks	110 depressed patients	”A significant decrease in BDI score (17.39–9.1) compared to the placebo (18.18–15.55) and prebiotic (19.72–14.14) supplementation (*p* = 0.042)”.
Romijin et al., 2017 [54]	Humans	8 weeks	79 participants (10 dropouts) with at least moderate scores on self-report mood measures	No significant impact on any psychological outcome measure.
Messaoudi et al., 2011 [50]	Animals (rats) and humans	4 weeks	66 participants based on a score of ≤12 in the HADS-anxiety subscale (HADS-A) and the HADS-depression subscale (HADS-D) and equal to or less than 20 in the HADS total score on the initial examination	A significant reduction in anxiety-like behavior in rats (*p* < 0.05) and alleviated psychological distress in volunteers, as measured particularly by the HSCL-90 scale, HADS, and CCL.
Pinto-Sanchez et al., 2017 [56]	Humans	6 weeks	44 adults with IBS and diarrhea or a mixed-stool pattern (based on the Rome III criteria) and mild-to-moderate anxiety and/or depression (based on the Hospital Anxiety and Depression Scale)	14 of 22 patients in the BL (Bifidobacterium longum) group had a reduction in depression scores of 2 points or more on the Hospital Anxiety and Depression Scale vs. 7 of 22 patients in the placebo group (*p* = 0.04). BL had no significant effect on anxiety or IBS symptoms.
Arseneault-Bréard et al., 2012 [51]	Animals (rats)	-	-	”Probiotics reversed the behavioural effects of myocardial infarction (MI) (*p* < 0.05), but did not alter the behaviour of sham rats. Intestinal permeability was increased in MI rats and reversed by probiotics. In conclusion, L. helveticus R0052 and B. longum R0175 combination interferes with the development of post-MI depressive behaviour and restores intestinal barrier integrity in MI rats.”
Diop et al., 2008 [55]	Humans	3 weeks	75 healthy volunteers with symptoms of stress	Significant reduction in 2 stress-induced gastrointestinal symptoms (abdominal pain and nausea/vomiting) and no significant modification of the other physical and psychological symptoms and sleep problems.
Gilbert et al., 2012 [52]	Animals (rats)	-	-	”administration of probiotics, starting after the onset of reperfusion, are beneficial to attenuate apoptosis in the limbic system and post-MI depression in the rat”.

## Data Availability

Data sharing is not applicable.

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
