# Peer review of "The Influence of Probiotic Supplementation on the Severity of Anxiety and Depressive Symptoms; Function and Composition of Gut Microbiota; and Metabolic, Inflammation, and Oxidative Stress Markers in Patients with Depression—A Study Protocol"

_metabolites, 2023, doi:10.3390/metabo13020182_

Round 1

Reviewer 1 Report (Previous Reviewer 1)

The article is interesting, but some suggestions are still needed:

The authors say (lines 69-70): “Experimental studies conducted in animal models in recent years have confirmed the relationship between dysbiosis (imbalanced microbiome) and depression-like behavior [18].” – However, they cite only 1 study.

- The tables in general are misconfigured and are not defined with the text;

- Figure 1 is not good. Figure 1 is out of shape and was not interesting for the article;

- Line 243 contains two commas (,,)

- Are the protocols used by the authors in the public domain? Please show this question. Otherwise, it is important to request the authorization of the authors.

Author Response

Reviewer 2 Report (Previous Reviewer 3)

The authors will conduct an interesting and important study. They plan to investigate the impact of Lactobacillus helveticus Rosell and Bifidobacterium longum Rosell (vs. placebo) over 8 wks in 100 patients with diagnosed depressive disorders. The authors assume a reduction of 2.9 in the MADRS score between both groups.

The reviewer raises important concerns that should be addressed before publication:

1. The authors are advised to discuss this study: J. Clin. Med. 2021, 10(7), 1342; https://doi.org/10.3390/jcm10071342. For the reviewer it is not clear why the authors report a second trial that seems to be already part of the previous protocol:

·       „PRO-DMS: probiotic + depression + MetS

·       PLC-DMS: placebo + depression + MetS

·       PRO-D: probiotic + depression

·       PLC-D: placebo + depression

2. Abstract: The primary endpoint is not clear. Please indicate that the effect on MADRS score will be studied and clearly distinguish between primary and secondary goals/endpoints.

3. Introduction 128 ff.: The authors should define primary and secondary objectives/endpoints.

4. 2.3. Statistical analysis 226 ff.: The authors list different methods that will be applied. In the opinion of the reviewer, it is important to refer to the sample size calculation and the primary endpoint (reduction in MADRS)

5. Results: Please check the numeration.

6. 3.5.1. Sample size calculation 256 ff.: In the opinion of the reviewer, a sample size of 100 is very ambitious to detect a significant difference between both groups. The authors should discuss the risk of bias/confounding/type II error by adding a separate limitations section.

7. 3.3. Interventions 327 ff.: The authors provided an overview about previous studies (depressive and anxiety symptoms studies) – Please harmonize this overview with the mentioned referenced in this section (e.g. ref. 47-50 and 51-53)

8. Table depressive and anxiety symptoms studies: Please extend this overview by adding further information (e.g. highlighting animal and human studies, trial design, duration, population etc.) – This table should give the rationale for conducting this specific trial. Furthermore, this table should be integrated into the main manuscript.

9. 3.4. Outcome measures 349 ff.: The sample size was calculated for a single endpoint (MADRS) – in this section and Table 3, the authors report various endpoints. Therefore, the authors are advised to define a single endpoint or adjust the alpha-level. The latter one will result in a higher sample size.

10. Non-published material: The authors should publish these documents as supplementary files.

Round 2

Reviewer 1 Report (Previous Reviewer 1)

The article must be accepted.

Reviewer 2 Report (Previous Reviewer 3)

Thank you for providing the revised version.

This manuscript is a resubmission of an earlier submission. The following is a list of the peer review reports and author responses from that submission.

Round 1

Reviewer 1 Report

The influence of probiotic supplementation on the severity of anxiety and depressive symptoms, as well as the function and composition of gut microbiota, metabolic, inflammation, and oxidative stress markers in patients with depression- a study protocol

The idea of ​​the article seems to be very interesting. The authors address current issues, but the article needs to be better structured. The submitted article looks much more like a research project than actually an original article.

The results item does not exist in the article. And the authors present tables that most resemble tables.

The authors put objectives in the article, since it is not necessary, since there is a justification at the end of the introduction.

The number of patients is also very small for the sample size. Serum parameters were evaluated as total cholesterol and fractions, but in fact I cannot see a direct relationship with the level of anxiety and depression. The ideal would be to quantify markers that could show a strong correlation with the assessed disorder. The body mass index was also evaluated, but this parameter is exceeded since it does not differentiate between the individual's fat mass and lean mass.

The article is outside the template.

Reviewer 2 Report

The study reported a protocol may use to evaluate the influence of probiotic supplementation on the severity of anxiety and depressive symptoms, as well as the function and composition of gut microbiota, metabolic, inflammation, and oxidative stress markers in patients with depression. However, there is no any clinical data showed in the draft. I was unable to give any comments to this draft. Please provide more sensible information. I do not recommend it.

Reviewer 3 Report

Article Type: In my opinion, “Article” should be replaced by “Study Protocol”

Introduction/Rationale: Please explain why the authors would use specifically Lactobacillus helveticus Rosell and Bifidobacterium longum Rosell. Also explain the rationale of intervene for 8 weeks.

Study groups:diagnosed depressive disorders” – need more specification

Design:

The study will be double blinded for the researchers, patients, and those who will analyze the resultsHow can the authors/study team guarantee a double-blind situation. Please specify the ordering/manufacturing process of both arms; are they supplied by the hospital pharmacy, ordered directly from the manufacturer (in this case indicate manufacturer and no.), who one distributes the pills, are the pill boxes labeled the same way, does only the pharmacy have the identification list etc. This paragraph needs much more content!

Randomization: Please specify this process. How different confounders / variables that might influence effect of intervention will be considered (BMI, heart failure, taking medications etc.)

Compliance: How compliance will be assessed? Please provide a sheet (Suppl. Material) that will be used by the patients to document pill intake.

Scale and questionnaires: Please provide all scales and questionnaires to suppl. material.

Objectives: For me it is not clear what are the primary and secondary endpoints.

Same from the introduction: “We plan to investigate the impact of the specific bacteria species on patients with depressive disorders with possible comorbid anxiety.

Please exactly define a primary endpoint and present a power analysis and your sample size calculation. Please directly refer to previous studies and their respective effect sizes to validate your assumed size of 50 + 50. Please provide a complete list (supplementary material) with specific endpoints and exact effect sizes of previous animal studies and human studies that prompted the authors to study the effect of Lactobacillus helveticus Rosell and Bifidobacterium longum Rosell.